# Comparison of Different d-SPE Sorbent Performances Based on Quick, Easy, Cheap, Effective, Rugged, and Safe (QuEChERS) Methodology for Multiresidue Pesticide Analyses in Rapeseeds

**DOI:** 10.3390/molecules26216727

**Published:** 2021-11-06

**Authors:** Saida Belarbi, Martin Vivier, Wafa Zaghouani, Aude De Sloovere, Valerie Agasse, Pascal Cardinael

**Affiliations:** 1Laboratoire SMS-EA3233, Normandie University, FR3038 INC3M, Unirouen, Place Emile Blondel, F-76821 Mont-Saint-Aignan, France; valerie.agasse@univ-rouen.fr; 2SGS France Laboratoire de Rouen, Technopôle du Madrillet, 65 Avenue Ettore Bugatti, BP 90014, F-76801 Saint Etienne du Rouvray, France; martin.vivier@sgs.com (M.V.); wafa.zaghouani@sgs.com (W.Z.); aude.desloovere@sgs.com (A.D.S.)

**Keywords:** multiresidue pesticide analyses, rapeseeds, QuEChERS extraction, dispersive solid-phase extraction, HPLC-MS/MS

## Abstract

Pesticide extraction in rapeseed samples remains a great analytical challenge due to the complexity of the matrix, which contains proteins, fatty acids, high amounts of triglycerides and cellulosic fibers. An HPLC-MS/MS method was developed for the quantification of 179 pesticides in rapeseeds. The performances of the quick, easy, cheap, effective, rugged, and safe (QuEChERS) method were evaluated using different dispersive solid-phase extraction (d-SPE) sorbents containing common octadecylsilane silica/primary–secondary amine adsorbent (PSA/C18) and new commercialized d-SPE materials dedicated to fatty matrices (Z-Sep, Z-Sep^+^, and EMR-Lipid). The analytical performances of these different sorbents were compared according to the SANTE/12682/2019 document. The best results were obtained using EMR-Lipid in terms of pesticide average recoveries (103 and 70 of the 179 targeted pesticides exhibited recoveries within 70–120% and 30–70%, respectively, with low RSD values). Moreover, the limits of quantification (LOQ) range from 1.72 µg/kg to 6.39 µg/kg for 173 of the pesticides. Only the recovery for tralkoxydim at 10 μg/kg level was not satisfactory (29%). The matrix effect was evaluated and proved to be limited between −50% and 50% for 169 pesticides with this EMR-Lipid and freezing. GC-Orbitrap analyses confirmed the best efficiency of the EMR-Lipid sorbent for the purification of rapeseeds.

## 1. Introduction

Rapeseeds are the second most important oilseed crops worldwide and are mainly grown in Eastern Europe and Asia. They contain functional compounds, such as dietary fibers, unsaturated fatty acids, proteins, and phenolic compounds, which are associated with several health benefits [1]. A wide range of insecticides (pyrethroids, organochlorines, organophosphorus, and carbamates), fungicides (phthalimides, triazoles, and imidazole sulfamides), and herbicides (sulfonylureas and diphenyl ethers) are used to increase harvest yields [2]. Some pesticides may remain in the seeds at high levels after harvest. These pesticides pose a serious threat to environmental and human health [3]. Between 2013 and 2020, several studies confirmed the presence of these pesticides in various fatty matrices, such as olive oil [4,5,6], soybeans [7], and sunflower oil [8].

To ensure consumer security, European authorities have established strict maximum residual limit (MRL) values for several pesticides. These MRL values can be very low; for instance, the MRL value was 0.01 mg/kg diflufenican in rapeseeds. The most frequently detected molecules in rapeseeds are pirimiphos-methyl, dichlorvos, and malathion. These insecticide residues are found at relatively low levels, on average from 0.1 to 0.25 mg/kg, in most crude oil samples from rapeseeds and sunflower seeds (70–80% of cases) and in the seeds themselves (20–30% of cases). Moreover, maximum levels of the sums of these residues in the crude oils can increase to 1 mg/kg [9].

To ensure the accurate quantification of pesticides in rapeseed samples, purification steps were necessary before injecting extracts to remove fat and coextracted compounds and to avoid matrix effects [10,11]. Indeed, rapeseed samples contain up to 40% fat, fibers, and proteins, and the major components are fatty acids (oleic, linoleic, and α-linoleic) and triglycerides [12]. Analysis of a wide range of pesticides at trace levels in rapeseed samples is very challenging because lipid coextracts can affect the extraction and quantification performances [13]. Moreover, non-polar and fat-soluble pesticides remain difficult to extract, leading to poor recoveries.

Several purification methods have been developed to eliminate the fatty matrix from the extracts of various oil seeds. Freezing-out is the simplest method for fat removal from the extract because fat precipitates below 0 °C and subsequently can be separated by centrifugation. Unfortunately, this method is time-consuming and does not totally remove the fatty matrix [14,15]. Gel permeation chromatography helps to separate low molecular mass compounds, such as pesticides, from high molecular mass compounds, such as lipids. However, this technique is not suitable for pesticides that have a high molecular weight, such as pyrethroids [16,17].

QuEChERS (quick, easy, cheap, effective, rugged, and safe) is the reference method for the extraction of pesticides in food matrices (fruits and vegetables) to achieve satisfactory recoveries [18]. This method was applied to the extraction of pesticide residues in various fatty matrices, such as vegetables containing high oil amounts, oils, and oilseeds, using different sorbents for d-SPE clean-up, such as PSA [19,20] and PSA/C18 [4,21]. This method appeared to be more convenient for enhancing pesticide recoveries and purifying extracts than the methods previously described (freezing and gel permeation). However, the extraction of some lipophilic pesticides remains problematic, especially in rapeseed. Currently, no sample preparation can sufficiently minimize the matrix effects for these specific compounds [22,23].

A new type of d-SPE sorbent was developed by Merck based on zirconia sorbent materials (Z-Sep, Z-Sep^+^) for fatty acid removal from fatty samples to improve matrix clean-up. Z-Sep is a silica support coated with zirconium dioxide. Distinct classes of active sites are present on its surface: Lewis acid sites, Brønsted acid sites, and Brønsted base sites. Z-Sep^+^ consists of silica particles coated with a zirconium dioxide layer, which is grafted with octadecylsilane groups. Zirconium dioxide material has been demonstrated to be a good adsorbent for carboxylic acids derived from fatty acids due to Lewis acid-base interactions [24]. The efficiency of this material has been demonstrated for pesticide residue quantifications in different vegetable oils [9] as well as in almonds and avocados [25].

Recently, studies with new d-SPE materials, such as the novel sorbent material Agilent Bond Eluant Enhanced Matrix Removal Lipid (EMR-Lipid), were published, demonstrating the selective adsorption of fat compounds without retention of targeted pesticides [26,27,28,29]. EMR-Lipid material is a porous sorbent that selectively retains long unbranched hydrocarbons that are characteristic of fatty matrices. Thus, large analytes such as pesticides cannot interact with the sorbent and remain in solution. QuEChERS extraction followed by EMR-Lipid d-SPE was successfully applied for multiresidue analyses of pesticides in avocado by GC-MS/MS [27] and LC-MS/MS [28]. The performance of EMR-Lipid has also been tested for other representative high lipidic matrices [29], including bovine liver [30], salmon [31], smoked fatty products of animal origin [32], and edible oils [33].

The main objective of this work was to evaluate different d-SPE materials, including Z-Sep, Z-Sep^+^, EMR-Lipid, and PSA/C18, as QuEChERS purification materials for pesticide analyses in rapeseeds. This study reported for the first time a comparison of the clean-up performances of these four sorbents for rapeseed extracts. For this, a sensitive, robust, and reliable multiresidue analytical method based on QuEChERS followed by LC-MS/MS was developed on 179 pesticides of various polarities and chemical families. Organic rapeseed samples were spiked at two levels, 10 µg/kg and 50 µg/kg. Then, the recovery rates, matrix effects, and LOQs were compared. Finally, the purified extracts were injected using GC-Orbitrap to estimate the amount of residual matrix compounds remaining and the purification efficiency.

## 2. Results and Discussion

An HPLC-MS/MS method was developed for the identification and quantification of 179 pesticides from a wide range of chemical classes of compounds (carbamates, chloroacetamides, benzamides, triazines, uracils, etc.) in rapeseed samples (Figure 1).

The sample preparation steps were carried out using the QuEChERS method. For the purification step (d-SPE), several sorbents were tested. The first sorbent selected was the PSA/C18 mixture, which is the reference mixture for fatty matrix purifications. The performance of the PSA/C18 mixture was compared with those of the new sorbents, EMR-Lipid, Z-Sep, and Z-Sep^+^, which were developed specifically for fatty matrix purification. Performances were evaluated in terms of recovery rates, repeatability, LODs, LOQs, and matrix effects. The purified extracts were also analyzed using a nontarget GC-Orbitrap method to identify possible remaining interferents.

### 2.1. HPLC-MS/MS Method Optimization

Chromatographic performances were satisfactory using methanol as an organic modifier in terms of retention, selectivity, and peak shape. The mobile phase pH also plays an important role in acid pesticide retention. Formic acid and ammonium formate were added to the mobile phase at 10 mM to adjust the pH, and they improved the chromatographic separation for these pesticides. The flow rate was set at 0.4 mL/min as a compromise between analysis time and resolution.

To achieve the best signal intensity in HPLC-MS/MS, the multiple reaction monitoring (MRM) transitions were optimized using individual standard solutions (1000 μg/L) in flow injection analysis mode. For electrospray ionization, positive mode (ESI^+^) was used for all pesticides. The precursor ions [M + H]^+^ of each pesticide were selected when they corresponded to the highest peak intensity. Otherwise, the adduct [M + NH_4_]^+^ (cyflumetofen) was chosen. After fragmentation, the transition of the highest sensitivity was used for the quantifier transition, and the second most sensitive transition was used for the qualifier transition. The detailed MRM transitions of each pesticide are listed in Appendix A Appendix A. All transitions were specific for each analyte, so the chromatographic resolution of pesticides is not mandatory, allowing the analytical run time to be reduced to 11.83 min.

### 2.2. Linearity

At concentrations ranging from 1 to 100 µg/L (1, 2, 4, 40, and 100 µg/L), the detector response was quadratic with a weighting of 1/x for almost all target molecules. A good correlation coefficient (more than 0.995) was observed for all the analytes. The S/N ratios of chromatographic peaks obtained for the lowest concentration of 1 µg/L were greater than 103 for both the quantifier and qualifier transitions. Residual values were calculated and showed a deviation of less than ±20% from the calibration curve for each calibration level. The standard deviation of the IS peak area was less than ±20%.

### 2.3. Recoveries and Precision

All recovery experiments were performed by analyzing rapeseed samples spiked a concentration of 10 µg/kg (*n* = 5) and 50 µg/kg (*n* = 5). The extraction recoveries of 179 pesticides, as well as the RSD (%), are presented in Appendix A. Three QC (quality control) levels (1, 2, and 4 µg/L) were injected after five sample injections to check the stability of the system. The standard deviations of the QC peak area were less than ±20%. As suggested by the SANTE/12682/2019 [34] document, the acceptable criteria concerning recoveries of pesticides should be within the range of 70–120%, with an associated repeatability RSD ≤ 20%. Moreover, mean recovery rates inside the range of 30–140% can be accepted if they are consistent, but the LOQ should be corrected.

For the majority of pesticides, the recoveries were similar for the samples spiked at 10 µg/kg and those spiked at 50 µg/kg. No recovery below 30% was observed with samples spiked at 50 µg/kg regardless of the purification sorbent used. For the samples spiked at 10 µg/kg, recoveries below 30% were obtained for a few molecules, specifically for the extracts purified with PSA/C18 sorbent. Because the same sorbent amounts were used for each spiking level, adsorption on the sorbent had a greater impact on the lower concentration of molecules. The results showed that extracts purified with EMR-Lipid sorbent exhibited better recoveries with 103 and 70 of the 179 targeted pesticides, which had recoveries within 70–120% and 30–70%, respectively, with low RSD values (Figure 2).

The recoveries were outside of the validation criteria for only six pesticides using this sorbent. The recovery of tralkoxydim was 29%, which might be explained by the specific interaction of this compound with polar moieties of the EMR-Lipid sorbent. For five pesticides (spirotetramat, saflufenacyl, isoxaflutole, foramsulfuron, and flazasulfuron), recoveries were up to 120%, demonstrating an overestimation due to a lack of purification efficiency.

In comparison with EMR-Lipid, recoveries obtained using Z-Sep and PSA/C18 sorbents exhibited generally lower values. Moreover, the recoveries were below 30% for 20 pesticides and 7 pesticides, demonstrating important interactions between these analytes and these sorbents. However, the use of Z-Sep^+^ sorbent enhanced the recoveries of some pesticides. A freezing step before PSA/C18 purification improved the recoveries compared to those obtained with single PSA/C18 purification. This result could be attributed to the precipitation of a lipid that released pesticides in the solvent.

The relationship between Log P and the recoveries of pesticides in rapeseed extracts spiked at 10 µg/kg depending on purification sorbents is presented in Figure 3.

For pesticides possessing a Log P < 2, the recoveries were below 60% for most of these pesticides when PSA/C18 and Z-Sep were used, confirming the strong polar interactions between the sorbent and polar analytes. These results were consistent with the polar surface of amino groups (PSA) and zirconium dioxide (Lewis acid). By contrast, the recoveries obtained using Z-Sep^+^ sorbent for the same kind of pesticide were higher due to the presence of octadecylsilane chains, which limited the accessibility of the zirconium dioxide surface.

It is interesting to note that 11 (cinosulfuron, fensulfothion, fensulfothion-oxon, fensulfothion-sulfone, fenthion sulfone, fenthion sulfoxide, flazasulfuron, foramsulfuron, florasulam, oxycarboxin, and rimsulfuron) of the 20 molecules with the lowest recovery (<30%) for the extracts purified with PSA/C18 have a sulfoxide or sulfone functional group in their structure. Moreover, the low recovery observed for ametoctradin could be related to the presence of an aliphatic hydrocarbon chain in its structure that can interact with the octadecylsilane chain grafted onto silica through hydrophobic interactions. It is important to note that a few compounds, such as oxycarboxin, presented low recovery values in unpurified and all purified extracts. In this case, the low recoveries could be attributed to the extraction step due to the high affinity of this compound to water.

### 2.4. Limits of Quantification

The document SANTE/12682/2019 [34] describes the LOQ as the minimum quantifiable concentration; the criteria for this value are a mean recovery within the 70–120% range and an RSD of <20%. Recovery rates outside the range of 70–120% can be accepted if they are consistent (RSD ≤ 20%), but the mean recovery should not be lower than 30% or above 140%. However, in these cases, a correction for recovery is required. Appendix A presents the LOD (µg/kg), LOQ (µg/kg) and matrix effects (ME%) for the 179 pesticides. For extracts purified with EMR-Lipid at a level of 10 µg/kg, only five molecules could not be quantified due to having recoveries outside of the performance criteria. Otherwise, the calculated LOQ values of the other pesticides were less than 10 µg/kg (from 1.72 µg/kg to 6.39 µg/kg), which is very satisfactory because they were lower than the MRL values. The method could not be validated at 10 µg/kg for foramsulfuron, isoxaflutole, saflufenacyl, spirotetramat, and tralkoxydim but could be validated at 50 µg/kg. The LOQs calculated for the pesticides in the extracts purified with the other sorbents showed that the values were on the same order of magnitude as those obtained with the EMR-Lipid sorbent, but the LOQs for 20, 7, and 6 pesticides could not be established using PSA/C18, Z-Sep, and Z-Sep^+^, respectively.

### 2.5. Matrix Effects

As a consequence of coeluting sample components, the targeted analyte signal may be enhanced or suppressed compared to the signal from the same targeted analyte when injected into a pure solvent. Matrix effects are evaluated by comparing the slope of the calibration curves for the standards in solvent against standards prepared in matrix extracts. The matrix effect (ME%) is calculated using Equation (1):Matrix effect (ME%) = ((slope of matrix/slope of solvent) − 1) × 100(1)

The soft matrix effect (suppression or enhancement of 0–20%) is negligible. However, if some of the analytes had a suppression or enhancement of 20–50%, the matrix effect appeared as medium. When the matrix effect (suppression or enhancement > 50%) is strong, it is necessary to use certain methods to overcome the ME, such as employing a matrix-matched calibration or sample dilution. The ME% values are presented in Appendix A.

Figure 4 shows that EMR-Lipid and Z-Sep presented similar results.

For rapeseed extract purified with EMR-Lipids, more than 90 molecules had a negligible matrix effect, and more than 70 pesticides had a medium matrix effect; these results confirm the efficiency of purification with the EMR-Lipid sorbent. The freezing process allowed us to significantly eliminate the matrix effect. Therefore, a combination of freezing followed by EMR-Lipid should be very efficient in limiting matrix effects. Moreover, the worst ME% values were obtained on the extracts of rapeseed purified with Z-Sep^+^, with more than 120 strong matrix effects. These purification materials appeared to be unsuitable for this specific matrix.

To evaluate the amount of the residual compounds in the extracts after the clean-up process, GC-Orbitrap analyses were performed in full scan mode. Rapeseed extracts purified with each sorbent in acetonitrile (ACN) were evaporated and reconstituted in a mixture of hexane/acetone (70/30, *v*/*v*). The chromatograms of the extracts are presented in Figure 5. The intensity of the total ion current obtained for the extract purified using EMR-Lipid (red line) was significantly less intense. The EMR-Lipid sorbent appeared to be very efficient in removing free fatty acids. The total ion currents observed for extracts purified with Z-Sep and Z-Sep^+^ presented a broad peak at approximately 10–14 min, demonstrating a lack of purification with these sorbents correlated to the strong matrix effect previously observed.

## 3. Materials and Methods

### 3.1. Chemicals and Reagents

Ultrapure water (18.2 MΩ·cm) was obtained from a Milli-Q water purification system (Millipore Ltd., Bedford, MA, USA). ACN and methanol (MeOH) were purchased from VWR (Fontenay-sous-Bois, France). Formic acid and ammonium formate were purchased from Sigma Aldrich (Saint Quentin Fallavier, France).

QuEChERS extraction kits were purchased from Agilent Technologies (Santa Clara, CA, USA): kits contained 4 g of magnesium sulfate (MgSO_4_), 1 g of sodium chloride (NaCl), 1 g of sodium citrate, and 0.5 g of sodium citrate sesquihydrate. PSA/C18 and EMR-Lipid sorbents were purchased from Agilent Technologies (Santa Clara, CA, USA). Z-Sep and Z-Sep^+^ sorbents were obtained from Supelco (Bellefonte, PA, USA).

High purity pesticide standards from a wide variety of chemical families and a large range of polarities (organophosphorus, carbamates, benzimidazoles, triazoles, pyridines, hydroxyanilides, strobilurins, etc.) were purchased from Sigma Aldrich (Steinheim, Germany) and the Dr. Ehrenstorfer Laboratory (Augsburg, Germany).

### 3.2. Preparation of Standard Solutions

Individual stock solutions were prepared by dissolving 10 mg of standards in 20 mL of ACN to obtain a solution of 0.5 g/L for each pesticide. An intermediate solution containing pesticides at 1000 µg/L was prepared by adding 0.1 mL to individual solutions to a 50 mL volumetric flask. Standard working solutions at various concentrations were prepared by dilution of the intermediate solutions in ACN appropriately. Then, a calibration range (1, 2, 4, 40, and 100 µg/L) was also prepared for the quantification step. Atrazine-d5 was also prepared at a concentration of 100 µg/mL, further diluted to 2 µg/mL in ACN, and added to the final concentration prior to HPLC-MS/MS analysis as an internal standard (IS). All stock and working solutions, including the IS, were stored in amber vials with Teflon-lined caps and then stored at −20 °C.

### 3.3. Samples and Spiking Procedure

Rapeseed samples were purchased from a local organic supermarket (Biocoop, Bois-Guillaume, France) and had been previously determined to be free of the target pesticides. All samples were mechanically ground until homogeneity was reached. For recovery studies, the organic rapeseed samples were spiked with the standard solution in ACN at 10 µg/kg and 50 µg/kg.

### 3.4. QuEChERS Method

A mass of 5 g of homogenized samples of rapeseed was weighed into a 50 mL disposable polypropylene centrifuge tube. Thereafter, ultrapure water (10 mL) was added, the mixture was stirred vigorously for one minute, 10 mL of ACN was added, and then, the mixture was immediately shaken for 1 min. Next, a salt mixture containing 4 g of anhydrous magnesium sulfate, 1 g of sodium chloride, 1 g of trisodium citrate dihydrate, and 0.5 g of disodium hydrogen citrate sesquihydrate was added for good separation of the water and ACN phases. The tubes were immediately shaken for 1 min and then centrifuged for 5 min at 4700 rpm at 20 °C. A volume of 6 mL of the supernatant was transferred into a polypropylene centrifuge tube containing various purification supports (QuEChERS d-SPE clean-up):(a)d-SPE with PSA/C18 sorbent (750 mg of PSA and 125 mg of C18)(b)d-SPE with EMR-Lipid sorbent (175 mg)(c)d-SPE with Z-Sep sorbent (175 mg)(d)d-SPE with Z-Sep^+^ sorbent (175 mg)

In addition, one sample of rapeseed extract was frozen at −20 °C before PSA/C18 purification to evaluate the influence of the freezing-out step. Purification tubes were immediately shaken for 1 min and then centrifuged for 5 min at 4700 rpm at 20 °C. A volume of 4 mL for each extract was collected and acidified with 40 µL of 5% formic acid in ACN. Finally, an unpurified rapeseed extract was also tested. The extracts were placed in a 1.5 mL vial (900 μL of extract and 100 μL of atrazine-d-5) and injected into the HPLC-MS/MS for the evaluation of the various sorbents.

### 3.5. HPLC-MS/MS Operating Conditions

HPLC-MS/MS using electrospray ionization in positive mode (ESI^+^) was used for the identification and quantification of the 179 targeted pesticides in rapeseed samples. A Thermo Scientific binary LC pump (Ultimate 3000 RS pump, LPG 3400 RS) equipped with an LC autosampler (Ultimate 3000 WPS-300 TRS) (Bremen, Germany) was operated at a flow rate of 0.4 mL/min using an Aqua^®^ C18 column 3 µm, 125 Å, 150 × 2.0 mm (Phenomenex, Torrance, CA, USA). A volume of 10 µL of the sample was injected. The binary mobile phase consisted of water with 0.2% (*v*/*v*) formic acid and 10 mM ammonium formate (phase A) and methanol with 0.2% (*v*/*v*) formic acid and 10 mM ammonium formate (phase B). The elution gradient started from 5% B (*v*/*v*) and was held for 0.6 min, increased to 64% B (*v*/*v*) at 2.4 min, and then increased to 90% B (*v*/*v*) at 5.4 min and held for 3 min. Then, the mobile composition was returned to the initial conditions over 0.8 min and was held for 2.63 min for re-equilibration. The total analysis time was 11.83 min.

A 5500 Q-TRAP (AB Sciex Instrument, Foster City, CA, USA) with an electrospray ionization source (ESI) was used for all experiments. The capillary voltage was maintained at 5500 V, and the temperature was set to 300 °C. Nitrogen was used at the nebulizer gas (GS1), auxiliary gas (GS2), and curtain gas (CUR) at pressures of 50, 60, and 26 psi. Argon was used as collision gas. For optimization of the MS/MS parameters of each pesticide, individual standard solutions were directly injected into the source. The declustering potential (DP), collision energy (CE), and collision cell exit potential (CXP) were automatically optimized by employing the automatic function of Analyst Software 1.6.3 (AB Sciex Instrument, Foster City, CA, USA). All pesticides were detected in the multiple reaction monitoring mode (MRM). Two MRM transitions (most sensitive) were selected, the first for quantification and the second for confirmation with a good ratio between them. The scheduled MRM mode with a time window of 60 s was selected for the detection of these molecules.

### 3.6. GC-Orbitrap Operating Conditions

Injections to evaluate the clean-up efficiency of the sample preparation step were performed using a GC-Q-Orbitrap system in full scan mode [35] (Q Exactive, Thermo Scientific, Bremen, Germany) consisting of a GERSTEL MPS (Multi-Purpose Sampler) (Mülheim, Germany) autosampler, a trace 1310 GC with a PTV injector, an electron ionization (EI) source, and a hybrid Q-Orbitrap mass spectrometer. A PTV Cool Injection System (CIS 6) was used with splitless mode injection (1 µL injected) with the following temperature program: at 0 of 60 °C with a hold time of 0.2 min, followed by a temperature increase at a rate of 720 °C/min until reaching 310 °C with a hold time of 5 min (run time: 20 min). Helium (99.999%) (Linde Gas, Schiedam, Netherlands) was used as a carrier gas at a constant flow of 1 mL/min. GC separations were performed using an HP-5 MS UI (30 m × 250 µm × 0.25 µm film thickness) (Agilent Technologies, Santa Clara, USA) column using the following temperature program: at 0 of 60 °C (1 min), a ramp up to 170 °C at 35 °C/min and then a subsequent increase to 310°C at a rate of 10 °C/min with a hold time of 2 min at 310 °C. The transfer line was maintained at 280 °C. Electron ionization was performed at 70 eV with the source temperature set at 280 °C. Full scan MS acquisition was performed in profile mode using an m/z range of 50–500. Nitrogen gas (Air liquid, Bagneux, France) was used for the C-Trap supply. For GC-Q-Orbitrap data processing, X-Calibur 4.0 (Thermo Scientific, Bremen, Germany) was used for peak identifications.

## 4. Conclusions

QuEChERS using ACN has proven to be an efficient method for extracting 178 pesticides (not including oxycarboxin) from rapeseeds. Among the d-SPE sorbents tested, the EMR-Lipid sorbent exhibited good performances in terms of recoveries, LOQs, and matrix effects. The rapeseed extracts purified with Z-Sep and PSA/C18 sorbents exhibited important interactions with polar analytes. Matrix effect values observed for the extracts purified with Z-Sep^+^ were not satisfactory, with more than 120 strong matrix effects. For some pesticides, better recoveries were obtained without clean-up, highlighting the adsorption of these molecules with some adsorbents. GC-Orbitrap analyses of the extract, performed in full scan mode, demonstrated that the EMR-Lipid sorbent was the most efficient for the elimination of fatty acids and lipids.

## Figures and Tables

**Figure 1 molecules-26-06727-f001:**
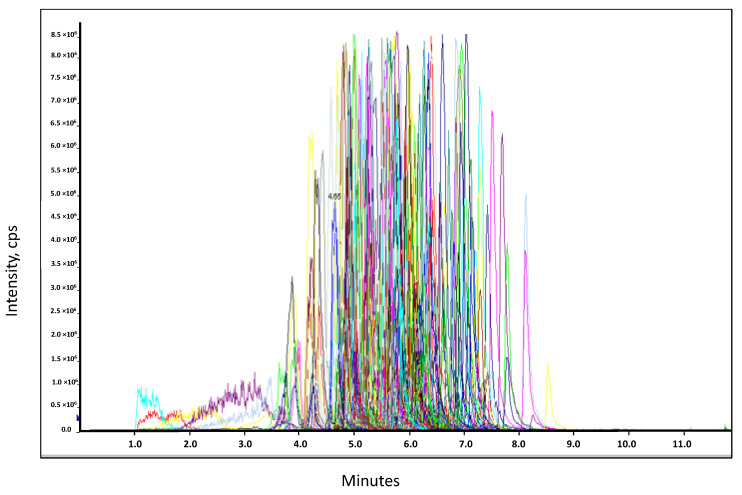
Extracted Ion Chromatograms (EIC) of the quantitative transition for the 179 pesticides analyzed in a rapeseed extract spiked at 10 µg/kg (each chromatogram was represented with different color).

**Figure 2 molecules-26-06727-f002:**
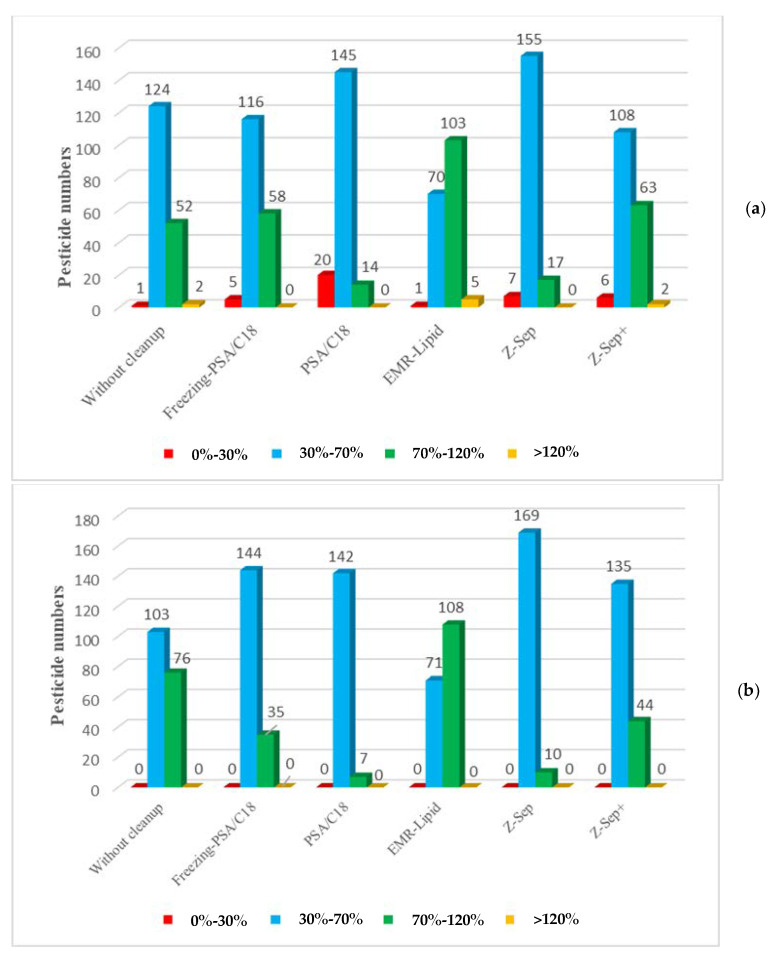
Pesticide recoveries in rapeseed extracts purified with different d-SPE sorbents at two spiking levels: 10 µg/kg (**a**) and 50 µg/kg (**b**).

**Figure 3 molecules-26-06727-f003:**
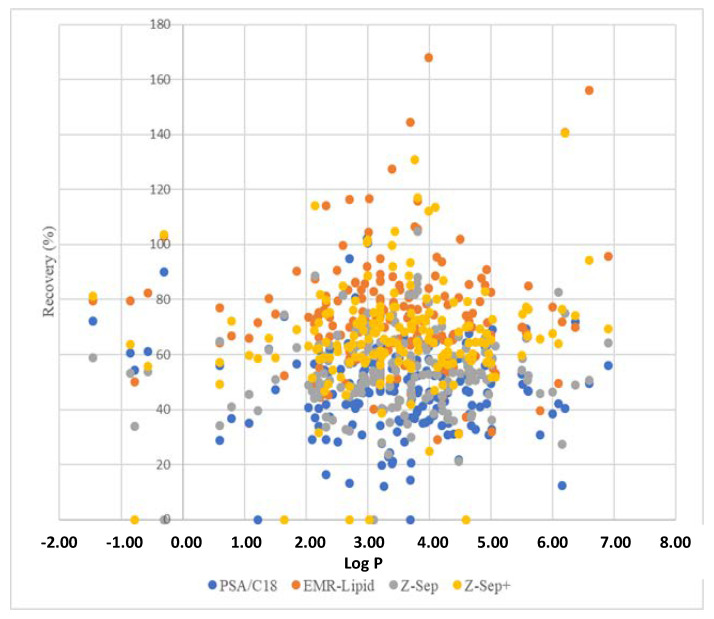
Relationship between Log P and pesticide recoveries from rapeseed samples using different purification adsorbents.

**Figure 4 molecules-26-06727-f004:**
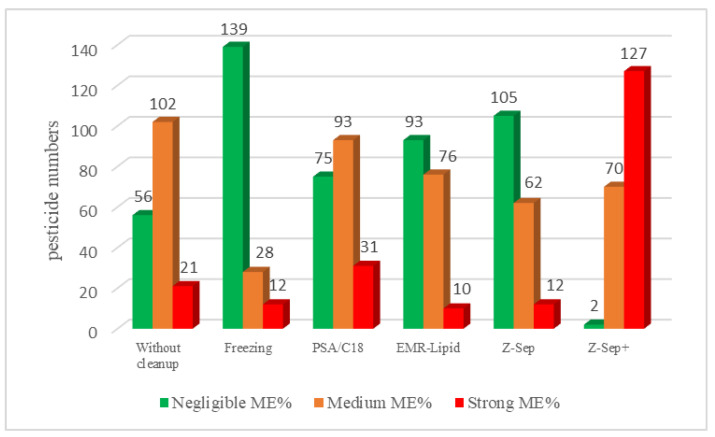
Matrix effect of pesticides in rapeseeds depending on the different purification sorbents.

**Figure 5 molecules-26-06727-f005:**
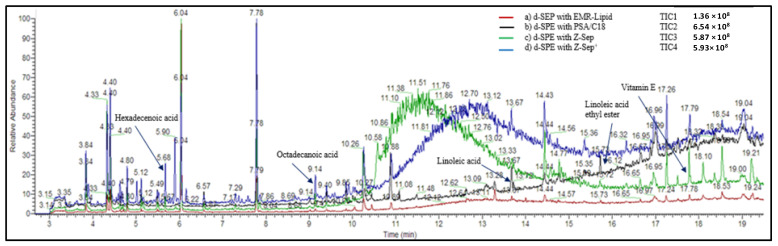
GC-Q-Orbitrap full scan chromatograms of rapeseed extracts using QuEChERS methodology with different d-SPE sorbents.

## Data Availability

The data presented in this study are available on request from the corresponding author.

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
