# Peer review of "Comparison of Different d-SPE Sorbent Performances Based on Quick, Easy, Cheap, Effective, Rugged, and Safe (QuEChERS) Methodology for Multiresidue Pesticide Analyses in Rapeseeds"

_molecules, 2021, doi:10.3390/molecules26216727_

Round 1
Reviewer 1 Report
Extraction of pesticides in rapeseed samples remains a great analytical challenge due to the complexity of the matrix. In this paper, a HPLC-MS/MS method was developed for the quantification of 179 pesticide residues in rapeseeds. The performance of QuEChERS method was evaluated by different purification methods, including freezing and applying four d-SPE sorbents containing PSA/C18, Z-Sep, Z-Sep+ and EMR-Lipid sorbents. Two spiking levels were tested in 5 replicates. The best results were obtained using EMR-Lipid in terms of pesticide average recoveries and LOQ for 173 pesticides. Only the recovery for tralkoxydim was not satisfactory. The matrix effect was evaluated and proved to be limited between -50% and 50% for 169 pesticides with this EMR-Lipid and freezing. GC-Orbitrap analyses confirmed the best efficiency of the EMR-Lipid sorbent for the purification of rapeseeds. This study can provide a scientific method for multiresidue pesticide analyses in plant seeds, but some problems need to be modified.
1.Abstract is the essence of this paper, which should include the key results of the research content, but some important results, such as the recovery rate data, are not reflected in the abstract, so it is suggested to improve this content.
2.The title of this paper is Comparison of different d-SPE sorbent performances based on QuEChERS methodology for multiresidue pesticide analyses in rapeseeds, However, it did not express the relevant content of QuEChERS. So please add this content.
- Page 9, line 348, “5” should be changed to “4”. Others similarly modify.
Author Response
- The authors agree to add more results in the abstract. So, the abstract was modified including the recovery rate data and partially reformulated to respect the maximum number of words (200).
- The title was modified as suggested (Quick, Easy, Cheap, Effective, Rugged and Safe).
- The number was changed.
Reviewer 2 Report
In the submitted manuscript – “Comparison of different d-SPE sorbent performances based on QuEChERS methodology for multiresidue pesticide analyses in rapeseeds” Belarbi et al. evaluate different d-SPE materials, including Z-Sep, Z-Sep+, EMR-Lipid and PSA/C18, as QuEChERS purification materials for pesticide analyses in rapeseeds. For this, a sensitive, robust, and reliable multiresidue analytical method based on QuEChERS followed by LC-MS/MS was developed on 179 pesticides of various polarities and chemical families.
To investigate 179 pesticides simultaneously is a very big challenge, however the authors solved it very well. It is not a question that these data are useful for scientist in this field. I recommend it for publication after some minor modification.
- HPLC-MS/MS chromatogram should provide in the article or in the Supplementary Information.
- Please supplement the Table S1 with retention times.
- Add the source of logP values (calculated or measured etc.)
- For electrospray ionization, positive mode (ESI+) was used 122 for all pesticides. Why? Based on the reviewer opinion for some pesticides negative ionization mode could be better.
- I think it would be useful to highlight that in some cases the method is better without cleanup than using a bad sorbent. It is clearly show the importance of the sorbent selection
- LINE 22 Only the recovery for tralkoxydim at 10 μg/kg level was not satisfactory (29%). Instead of Only the recovery for tralkoxydim was not satisfactory (29%).
- Line 140 recoveries instead of Recoveries
- Line 327 scheduled instead of Scheduled.
- Line 349 ACN instead of acetonitrile
Author Response
- A chromatogram (TIC) obtained by injecting an extract containing a mixture of 179 pesticides was added as figure 1.
- Retention times were added in table S1
- Log P values were obtained from PubChem (website).
- All pesticides were analyzed in ESI+ mode and were detected with low LOQ. The duty cycle needed to analyze the pesticides in both mode (ESI+ and ESI- in the same run) with our mass spectrometer is too long to perform analysis of 179 pesticide even if LOQ could be better for some molecules using ESI- mode.
- The following sentence was added in the conclusion to highlight this point: ‘In some cases the method is better without cleanup than using a bad sorbent. It is clearly shown the importance of the sorbent selection'
- The sentence was modified
- The mistake was corrected
- The mistake was corrected
- The mistake was corrected
Reviewer 3 Report
This manuscript compared the purification effects of several purification materials based on QuEChERS method, and found EMR-Lipid sorbents has the best recoveries and LOQ. Then GC-Orbitrap analyses confirmed the best efficiency of the EMR-Lipid sorbent for the purification of rapeseeds. This article is generally innovative, but the workload is relatively solid, which can provide a certain data basis for the development of related fields.
Some minor errors in the manuscript:
(1) A letter “b” is missing in figure 1b
(2) The innovation and importance of the article should be strengthened in the introduction section.
(3) The references of the article are generally older, and more references in recent five years should be cited.
Author Response
- The mistake was corrected
- The following sentence was added in the introduction to highlight the innovation. This study reported for the first time a comparison of the clean-up performances of these four sorbents for rapeseed extracts.
- Three recent references were added